# Exploring Gluconamide-Modified Silica Nanoparticles of Different Sizes as Effective Carriers for Antimicrobial Photodynamic Therapy

**DOI:** 10.3390/nano14241982

**Published:** 2024-12-11

**Authors:** Ruth Prieto-Montero, Lucia Herrera, Maite Tejón, Andrea Albaya, Jose Luis Chiara, Mónica L. Fanarraga, Virginia Martínez-Martínez

**Affiliations:** 1Departamento de Química Física, Facultad de Ciencia y Tecnología, Universidad del País Vasco, UPV-EHU, Apartado 644, 48080 Bilbao, Spain; lherrera009@ikasle.ehu.eus (L.H.); maite.tejon@ehu.eus (M.T.); andrea.albaya@csic.es (A.A.); 2Instituto de Química Orgánica General (IQOG-CSIC), Juan de la Cierva 3, 28006 Madrid, Spain; jl.chiara@csic.es; 3Grupo de Nanomedicina Instituto de Investigación Valdecilla-IDIVAL, Universidad de Cantabria, Herrera Oria s/n, 39011 Santander, Spain; monica.lopez@unican.es

**Keywords:** antimicrobial resistance, photosensitizer, photodynamic therapy, silica nanoparticles, Rose Bengal, Gram-negative bacteria, gluconamide, biotargeting

## Abstract

Antimicrobial resistance (AMR), a consequence of the ability of microorganisms, especially bacteria, to develop resistance against conventional antibiotics, hampering the treatment of common infections, is recognized as one of the most imperative health threats of this century. Antibacterial photodynamic therapy (aPDT) has emerged as a promising alternative strategy, utilizing photosensitizers activated by light to generate reactive oxygen species (ROS) that kill pathogens without inducing resistance. In this work, we synthesized silica nanoparticles (NPs) of different sizes (20 nm, 80 nm, and 250 nm) functionalized with the photosensitizer Rose Bengal (RB) and a gluconamide ligand, which targets Gram-negative bacteria, to assess their potential in aPDT. Comprehensive characterization, including dynamic light scattering (DLS) and photophysical analysis, confirmed the stability and effective singlet oxygen production of the functionalized nanoparticles. Although the surface loading density of Rose Bengal was constant at the nanoparticle external surface, RB loading (in mg/g nanoparticle) was size-dependent, decreasing with increasing nanoparticle diameter. Further, the spherical geometry of nanoparticles favored smaller nanoparticles for antibacterial PDT, as this maximizes the surface contact area with the bacteria wall, with the smallest (20 nm) and intermediate (80 nm) particles being more promising. Bacterial assays in *E. coli* revealed minimal dark toxicity and significant light-activated phototoxicity for the RB-loaded nanoparticles. The addition of gluconamide notably enhanced phototoxic activity, particularly in the smallest nanoparticles (RB-G-20@SiNP), which demonstrated the highest phototoxicity-to-cytotoxicity ratio. These findings indicate that small, gluconamide-functionalized silica nanoparticles are highly effective for targeted aPDT, offering a robust strategy to combat AMR.

## 1. Introduction

Antimicrobial resistance (AMR) is a growing global health threat, underscored by the staggering statistic of 1.27 million annual deaths worldwide in 2019 directly attributed to infections caused by resistant microorganisms and nearly 5 million deaths associated with resistant infections [1,2,3,4]. The WHO further estimates that AMR could claim up to 10 million lives per year by 2050, and World Bank indicates it could lead to cumulative healthcare expenses of up to USD 1 trillion by that time [5,6]. AMR arises from the evolution of microorganisms, primarily Gram-negative bacteria that have acquired the ability to resist the effects of antibiotics. This resistance is often mediated by genetic changes, such as mutations or the acquisition of new genetic material, which enable bacteria to modify their structure or components, rendering them less susceptible to antibiotic action [1,7,8,9,10]. Although new antibacterials are being developed, only a few of them can be considered as a new type. Further, their implementation in the market is slow and their accessibility is limited, not reaching worldwide patients. Thus, nontraditional agents and treatments have to be explored as complements to antibiotics, from which the microorganism cannot develop resistance, to mitigate AMR.

Antimicrobial photodynamic therapy (aPDT) offers a promising alternative to conventional antibiotic treatments in the face of rising AMR. This therapeutic approach involves the use of a photosensitizer (PS), a light-sensitive molecule, in conjunction with light of a specific wavelength and intensity. Upon activation, the PS generates highly reactive oxygen species (ROS), primarily singlet oxygen (^1^O_2_). These short-lived ROS species can effectively destroy cells, viruses, and bacteria, both directly and by disrupting their ability to transfer resistance genes to other microorganisms [11,12,13,14,15,16,17,18,19]. This rapid and localized action of aPDT helps to mitigate the spread of resistance, making it a valuable tool in combating the global AMR crisis. As the action in PDT is rapid and local, it is crucial to precisely localize the PS in the infected area. However, most of the available PSs (porphyrins, cyanins, xanthenes, BODIPYs, etc.) [18,19,20,21,22] do not present any special affinity for bacteria, being also accumulated inside normal cells, potentially causing side effects [18,20]. In addition, the hydrophobic nature of many photosensitizers limits their solubility in aqueous media, leading to the formation of molecular aggregates that considerably reduce the phototoxic action and, consequently, the effectiveness of PS.

In recent years, the use of nanoparticles (NP) has brought an important impact allied with overcoming the inherent PS limitations. The rational functionalization of the nanoparticle surface with targeting ligands transforms them into active vehicles for PS, increasing its selectivity, stability, biofilm penetration, solubility, and bioavailability in physiological media [23,24,25,26,27,28,29,30,31]. Nowadays, several types of nanoparticles, mainly lipid-based, are approved for their clinical use [32], and many others (polymeric and inorganic) are in (pre)clinical trials to evaluate their effectiveness and safety. This is the case for silica nanoparticles, which have entered clinical trials for a variety of biomedical applications [28,31,33,34,35,36]. Silica nanoparticles offer several advantages compared to organic nanoparticles because of their excellent physicochemical properties (chemically inert, mechanically stable, and optically transparent), tunable structure (particle size and morphology and pore diameter), and customizable surface functionality (as a consequence of a high presence of silanol groups on the surface) [37,38,39,40,41,42,43]. In this work, dense silica nanoparticles with different diameters (25 nm, 80 nm, and 250 nm) were synthesized by the Stöber or arginine method [44,45]. The silica nanoparticles are externally functionalized with a commercial photosensitizer, Rose Bengal (RB), a widely used commercial PS characterized by an intense absorption band in the green region (λ_abs_ = 520–580 nm), modest fluorescence (Φ_fl_ = 10%), and high singlet oxygen production (Φ_∆_ = 86%) [43,44]. In addition, a gluconamide derivative (G) (Figure 1) is used as a targeting ligand for Gram-negative bacteria. RB is used here as a standard photosensitizer to elucidate the effect of the particle size of the gluconamide-modified and unmodified silica nanoparticles on the toxicity under dark and irradiated conditions in *E. coli* bacteria. This open-chain carbohydrate has demonstrated specific adhesion by favorable interaction with the bacteria lipopolysaccharide LPS outer membrane [46]. Additionally, the gluconamide coating prevents protein corona formation [47] (a nonspecific protein adhesion, which modifies the physicochemical properties of nanoparticles’ surface and whose composition and extent depend on the nanoparticle type, biological medium, and exposure time) and consequently preserves its targeting ability [48]. Gluconamide on the external surface also reduces particle aggregation, increasing the stability of the nanosystem in the biological medium. The results of this study could be extended to other photosensitizers to expand the development of inexpensive, attainable, and efficient nanoplatforms for aPDT to combat AMR.

## 2. Materials and Methods

### 2.1. Materials

Tetraethoxysilane (TEOS, Sigma-Aldrich, St. Louis, MO, USA), ammonium hydroxide solution (28% NH_3_, Sigma-Aldrich), arginine (Sigma-Aldrich), Rose Bengal (RB, Sigma-Aldrich), ethyl chloroformate (Across), triethylamine (Sigma-Aldrich), (3-aminopropyl)trimethoxysilane (APTMS, Sigma-Aldrich), and *N*-(3-(triethoxysilyl)propyl)-d-gluconamide (“gluconamide”, Gelest, Morrisville, PA, USA) were obtained from the indicated commercial sources.

### 2.2. Silica Nanoparticle Synthesis

Synthesis of 20 nm silica NPs: to a solution of l-arginine (26.7 mg, 0.15 mmol) in H_2_O (25 mL), TEOS (4 mL, 18 mmol) was added, and the mixture was stirred at 60 °C for 72 h. The resulting 20@SiNPs were washed with ethanol and water by centrifugation (15,000 rpm = 21,130 rcf, 1 h) and then dried at 100 °C for 24 h.

Synthesis of 80 nm silica NPs: NH_4_OH (3.1 mL, 80 mmol) was added to a water–ethanol mixture (3.4 mL and 100 mL, respectively) and, after stirring for 30 min, TEOS was added (3.8 mL, 17 mmol). The mixture was stirred at 27 °C for 24 h. The resulting 80@SiNPs were washed with ethanol and water by centrifugation (15,000 rpm = 21,130 rcf, 30 min) and dried at 100 °C for 24 h.

Synthesis of 250 nm silica NPs: NH_4_OH (8.4 mL, 174 mmol) was added to the water–ethanol mixture (6.4 mL and 40 mL, respectively) and, after stirring for 30 min, TEOS was added (4.2 mL, 19 mmol). The mixture was stirred for 3 h at 58 °C. The resulting 250@SiNPs were washed with ethanol and water by centrifugation (15,000 rpm = 21,130 rcf, 30 min) and were dried at 100 °C for 24 h.

SiNP functionalization with Rose Bengal and gluconamide:

A solution of Rose Bengal (26.4 mg, 0.026 mmol) in anhydrous methanol (20 mL) was stirred under a nitrogen atmosphere at 0 °C. Ethyl chloroformate (3 μL, 0.026 mmol) and triethylamine (4 μL, 0.026 mmol) were added dropwise. After further stirring for 30 min, (3-aminopropyl)trimethoxysilane (5 μL) was added to the reaction mixture, which was further stirred for 1 h at 0 °C. Then, SiNP (30 mg) and *N*-(3-(triethoxysilyl)propyl)-d-gluconamide (11 μL, 0.026 mmol) were added and mixed for 1 h at room temperature. The resultant suspension of fully functionalized SiNP with RB and gluconamide (named RB-G-20@SiNP, RB-G-80@SiNP, and RB-G-250@SiNP, for 20, 80, or 250 nm NPs size) was centrifuged (15,000 rpm =21,130 rcf) and washed with ethanol until a colorless supernatant was obtained. The RB-G@SiNP was collected by filtration and was rinsed several times with water to ensure ethanol-free systems (see Appendix A). As a control sample, silica NPs functionalized only with RB, without gluconamide (named RB-20@SiNP, RB-80@SiNP, and RB-G250@SiNP, respectively), were also synthesized.

### 2.3. Structural and Chemical Characterization

The size, shape, and morphology of the obtained NPs were characterized by transmission electron microscopy (TEM) using a JEOL JEM 1400Plus operating at 100 kV. Images were acquired with an sCMOS Hamamatsu digital camera. Dynamic light scattering (DLS) and Zeta potential (ζ) measurements were performed in nanoparticle suspension (0.1 mg/mL) to analyze their hydrodynamic diameter and electrophoretic stability using a Malvern Zetasizer Nano ZS, which has a Helium-Neon (λ = 633 nm) laser. Elemental surface compositions (in atomic percentage, at%) of the silica nanoparticles were analyzed by X-ray photoelectron spectroscopy (XPS, SPECS equipment). The measurements were carried out by wide scan: energy step 0.1 eV, dwell time 0.1 s, and pass energy 30 eV with a 901 electron exit angle. FTIR spectra were obtained from powder samples using the ATR technique with an Affinity-1S Shimadzu spectrometer (4000–400 cm^−1^ range).

### 2.4. Photophysical Characterization

The absorption spectra were recorded with a UV-Vis-NIR spectrometer (model Cary 7000, Agilent Technologies, Madrid, Spain). In the case of the nanoparticle samples, an integrating sphere (model Internal DRA 900, Livingston, UK) was used for uniform light collection to correct the reflection and scattering effects of the samples. The fluorescence measurements were recorded with an Edinburgh Instruments Spectrofluorimeter (FLSP920 model, Livingston, UK) equipped with a 450 W xenon flash lamp as the excitation source. The fluorescence spectra were corrected from the wavelength dependence on the detector sensibility. The fluorescence quantum yields of the photosensitizers were measured by the relative method, using PM597 (Φ_fl_ = 0.48 in methanol) as the standard sample dye [49], following the equation:$$\Phi_{fl}=\Phi_{fl}^{R}\frac{\eta^{2}}{{\left(\eta^{R}\right)}^{2}}\frac{F}{F^{R}}\frac{{(1-10}^{A^{R}})}{\left(1-10^{A}\right)}$$

The singlet oxygen (^1^O_2_) production was determined by direct measurement of its phosphorescence at 1276 nm (Appendix A), employing an NIR detector (InGaAs detector, Hamamatsu G8605-23), integrated into the same Edinburgh spectrofluorimeter upon continuous monochromatic excitation (450 W Xenon lamp) of the sample. Singlet oxygen quantum yields (Φ_∆_^PS^) were calculated by the relative method, using commercial Rose Bengal (RB, Φ_∆_^PS^ = 0.86 in CD_3_OD) [50] as reference, following the equation:$$\Phi_{\Delta }^{PS}=\Phi_{\Delta }^{R}\frac{S_{e}^{PS}}{S_{e}^{R}}\frac{{(1-10}^{A^{R}})}{\left(1-10^{A^{PS}}\right)}$$

The amounts of RB attached to the external surface of silica NPs were estimated photometrically in RB@SiNPs stable suspensions, assuming the same molar extinction coefficients for the RB in solution and covalently attached to the silica surface.

### 2.5. Antibacterial Activity

The strain used in this study is *E. coli* BL21 (DE3), a derivative of the *E. coli* B strain. The protocol followed to expose the bacteria to the RB in solution or RB@SiNPs is depicted in Appendix A. Briefly, the bacterial precultures were regrown until they reached an optical density (OD) of 0.6. A total of 750 μL of these cultures was resuspended in PBS. For light exposure, 50 μL aliquots of this *E. coli* suspension were placed in the wells of a 24-well plate, followed by the addition of various solutions of RB or dispersions of PS@SiNP. To ensure an even distribution of the added compounds, the plate was vigorously shaken at 250 rpm for 30 min before irradiation. Irradiations were performed using light-emitting diode devices: LED Par 64 Short Q4-18 (Showtec, Burgebrach, The Netherlands) in the green region (wavelength centered at 518 nm, 10 mW/cm^2^), as shown in Appendix A.

The total light dosage for green light irradiation (TLD) was previously optimized for the RB system [51] by varying the exposition time, from 30 min (16 J/cm^2^) to 120 min (65 J/cm^2^), the total light dosage being set at 65 Jcm^−2^ and it was the one used in this work.

To assess bacterial viability post-treatment, *E. coli* bacterial suspensions were subjected to serial dilution, with 100 μL of each dilution plated onto LB agar plates (see Appendix A). These plates were then incubated overnight at 37 °C, after which the number of viable bacteria was determined by counting the colony-forming units (CFU). This count was compared to the untreated control group. Each condition was replicated three times, and each experiment was conducted on three separate days. Bacterial survival is expressed as the percentage relative to the control CFU count (under dark conditions). Statistical differences between the control (untreated bacteria in dark conditions) and treated *E. coli* were assessed using a *t*-test through the GraphPad Prism program version 8.01. Additional statistical differences between dark and light exposures at the same concentrations were also analyzed through a *t*-test through the GraphPad Prism program version 8.01. The significance level grade is *p* value: * < 0.033, ** < 0.002, *** < 0.0002, and **** < 0.0001, n = 3, DF.

### 2.6. Bacteria Imaging

For transmission electron microscopy (TEM) imaging of bacteria and nanoparticles, 150 μL of bacterial suspension, with OD = 0.6, was incubated with RB-G-20@SiNP, RB-G-80@SiNP, or RB-G-250@SiNP (0.15 mg/mL) in PBS. After 2 h of incubation, the bacteria were fixed in 4% paraformaldehyde for 20 min and washed with water four times. The samples were deposited on carbon-coated copper grids. Imaging was performed using Philips SuperTwin, CM200 (Thermo Fisher Scientific, Eindhoven, The Netherlands) at 200 kV, with LaB_6_ filament applying different magnifications.

## 3. Results and Discussion

### 3.1. Silica Nanoparticles Characterization

Silica nanoparticles with diameters of 20 nm, 80 nm, and 250 nm were selected to investigate the effect of particle size on antimicrobial photodynamic therapy (PDT). The success of these syntheses was confirmed by TEM and DLS (Figure 2 and Appendix A and Table 1). The three samples show spherical shapes with a narrow size distribution. According to the PDI index (Table 1), values ≤ 0.21 obtained for nanoparticles of 80 nm and 250 nm diameters indicate a narrower size distribution compared to the 20 nm particles, which have a slightly higher PDI value (~0.4). The average size distribution obtained from TEM (Figure 2 and Appendix A) is 21 ± 2 nm for the 20@SiNPs, 80 ± 5 nm for the 80@SiNPs, and 260 ± 27 nm for 250@SiNPs. These values are consistent with the hydrodynamic diameter measured for 80@SiNPs and 250@SiNPs (0.1 mg/mL water suspension) by DLS (Table 1). However, 20@SiNPs, which are the smallest in size, showed a larger hydrodynamic diameter, suggesting possible particle interaction in suspension. Regarding water stability, the ζ-potential values, higher than ±25 mV in the three samples, indicate good stabilities of the suspension in water and a low tendency to forming larger aggregates [52]. The surface composition of these nanomaterials was studied by XPS (Table 1 and Appendix A). XPS spectra show the characteristic signals of Si (2p) at 102.5 eV and O (1s) at 523 eV, with a relative ratio 1:2. The noisy signal assigned to C (1s) is of around 2–3% and is assigned to adventitious carbon.

### 3.2. Silica Nanoparticles Postfunctionalization

The nanoparticles with different sizes (20, 80, and 250 nm) were externally functionalized with Rose Bengal, affording RB-20@SiNP, RB-80@SiNP, and RB-250@SiNP, respectively. The nanosystems were further modified with the gluconamide ligand, yielding RB-G-20@SiNP, RB-G-80@SiNP, and RB-G-250@ SiNP, respectively (Appendix A). The sequential addition of RB and G induces changes in the ζ-potential values. The hydrophobic character of the organic RB dye decreases ζ-potential by approximately 10 mV (less negative values), which is practically restored after the subsequent introduction of the hydrophilic gluconamide derivative, ensuring good stability of the nanosystems in water (Table 2). In contrast, the hydrodynamic diameter is not significantly affected by the presence of RB and G on the external surface (Table 2). The covalent attachment of RB and G was confirmed by FTIR (Appendix A), showing the characteristic bands at 1635 cm^−1^ and 1550 cm^−1^ of the amide group, which are more evident in the fully functionalized samples with RB and G.

The amount of RB externally attached to the silica nanoparticles was quantified by the photometric method, based on the absorption spectra of the nanoparticle suspensions (Figure 3; Table 2). The total RB loading per gram of nanoparticles decreases with increasing particle size, consistent with the reduced surface-area-to-volume ratio (Appendix A). Specifically, RB-G-20@SiNP has a loading of 7 μmol/g, RB-G-80@SiNP has 2.1 μmol/g, and RB-G-250@SiNP has 0.8 μmol/g, which is an order of magnitude lower than that of RB-G-20@SiNP (Table 2). From these experimentally determined weight ratios, the surface density of RB molecules was calculated, yielding approximately 0.023, 0.027, and 0.029 molecules/nm^2^ for RB-G-20@SiNP, RB-G-80@SiNP, and RB-G-250@SiNP, respectively (Appendix A). These results confirm that the surface density of PS molecules is consistent across all particle sizes, independent of diameter.

Despite the constant surface density of RB molecules across nanoparticles of different sizes, photodynamic antibacterial activity is anticipated to vary depending on particle size under the experimental conditions where a fixed overall RB concentration was used (see below). Smaller nanoparticles have a higher surface-area-to-volume ratio, resulting in a higher number of particles per unit volume in suspension and a better surface accessibility for both dissolved molecular oxygen and the bacterial cell membrane. The increased number of nanoparticles interacting with the cell membrane can enhance the efficiency of ROS generation at the membrane surface, improving the overall antibacterial effect. Consequently, smaller nanoparticles, such as RB-G-20@SiNP and RB-G-80@SiNP, are expected to be more effective in antimicrobial photodynamic therapy (aPDT) than the larger RB-G-250@SiNP, even though all nanoparticles have the same surface density of RB molecules [53,54].

The absorption and emission spectra of the three completed nanosystems reveal very similar features to the RB in solution, in position and shape (Figure 3 and Table 3), indicating the absence of dye aggregation at the nanoparticle surface. Moreover, the photophysical properties in terms of fluorescent and singlet oxygen quantum yields are similar to RB in methanolic solution keeping a modest fluorescence efficiency (Φ_fl_ ≥ 0.05) and high singlet oxygen production (Φ_∆_ > 0.75) for RB-G-20@SiNP and RB-G-80@SiNP (Table 3). In particular, RB-G-20@SiNP demonstrated the best photophysical properties together with a higher PS loaded at the nanoparticle, suggesting this sample is the most promising for aPDT.

### 3.3. Antimicrobial Studies with Gram-Negative Bacteria

Illustrative TEM images of the three nanosystems, RB-G-20@SiNP, RB-G-80@SiNP, and RB-G-250@SiNP, incubated with *E. coli* bacteria were taken (Figure 4). TEM images reveal very good adhesion of the three gluconamide-decorated silica nanosystems to the external wall of the *E. coli* bacteria. Fewer nanoparticles are located on the bacterial outer membrane in the case of RB-G-250@SiNP due to their larger size, which probably hinders interaction with the external LPS layer of the bacterial membrane, as previously discussed (see the preceding section). This fact, combined with the previously mentioned that a higher nanoparticle concentration (mg/mL) is required to achieve the desired RB concentration, means this sample is discarded for the bio-assays in *E. coli*.

To evaluate the potential of functionalized silica nanoparticles as a novel platform for aPDT against Gram-negative bacteria, we investigated the cytotoxicity and phototoxicity of nanoparticles with varying sizes (20 nm and 80 nm) and surface modifications (with and without gluconamide targeting ligand). Our study aimed to compare the activity of these nanoparticle-based aPDT systems to that of the photosensitizer RB in solution. Additionally, we explored the influence of the gluconamide ligand on the phototoxic efficacy of the nanoparticles, seeking to understand how this modification might enhance their antibacterial activity. The viability studies of these nanoparticles in *E. coli* bacteria were carried out under dark and green irradiation. In our previous work, the protocol was optimized for RB in solution, setting the concentration at 5·10^−7^ m and light dose to 65 J/cm^2^ [51].

The four systems, RB-20@SiNP, RB-G-20@SiNP, RB-80@SiNP, and RB-G-80@SiNP, were tested in *E. coli* under dark conditions and green irradiation, and their performance was compared with free RB in solution (Figure 5). RB in solution displayed the highest phototoxicity, leading to the lowest % CFU (12%). However, this high phototoxicity is partly assisted by the cytotoxicity under dark conditions, which already leads to approximately 60% bacterial death. Although the reason for the RB intrinsic cytotoxicity in solution under dark conditions is not yet fully understood, previous studies showed that it increases with the incubation time or photosensitizer concentration, possibly as a consequence of the membrane permeabilization process by RB molecules [55,56].

A different scenario is observed for the photosensitized nanosystems (Figure 5). In dark conditions, the cytotoxicity of the four nanosystems is totally mitigated, with 96–99% CFU remaining viable, except for RB-20@SiNP, which induces a slight bacterial death (83% of CFU are alive). In these cases, RB is not able to cross the bacteria membrane as it is covalently linked to SiNPs and, accordingly, dark toxicity from RB is avoided.

The photoactivity of the four nanosystems, RB-20@SiNP, RB-G-20@SiNP, RB-80@SiNP, and RB-G-80@SiNP, resulted in a bacterial viability range of 25–50% CFU. There are significant differences for those systems functionalized with gluconamide moiety. For instance, RB-20@SiNP and RB-80@SiNP lead to 35% and 48% CFU when compared to the dark control, respectively. However, their homologous nanosystems with gluconamide, RB-G-20@SiNP and RB-G-80@SiNP, reach 27% and 34% CFU, respectively, which represent an increase in the phototoxicity under green irradiation when compared with those without the targeting ligand. Further, RB-G-20@SiNP has completely reduced the dark cytotoxicity, which was not the case for RB-20@SiNP.

These results confirm the critical role of gluconamide targeting on the nanoparticle surface in enhancing phototoxicity while preventing dark cytotoxicity. This effect is likely due to the improvement of the stability of nanoparticles in water avoiding aggregation, according to the ζ-potential results (Table 2), and the favorable interaction between nanoparticles and the bacterial cell wall, proposed in previous studies [46].

Regarding the size effect of the silica NPs, both the 20 nm and 80 nm nanosystems show promising results against Gram-negative bacteria, with RB-G-20@SiNP demonstrating slightly better performance under green light irradiation.

## 4. Conclusions

Silica nanoparticles of different sizes (20 nm, 80 nm, and 250 nm) functionalized with commercial RB as a photosensitizer and gluconamide as a specific targeting ligand for Gram-negative bacteria were synthesized and evaluated as candidates for antimicrobial photodynamic therapy (aPDT). Comprehensive characterization, including DLS measurements, confirmed the good stability of the nanosystems in aqueous media, particularly after covalent functionalization with gluconamide. The photophysical properties of RB attached to the silica nanoparticles remained comparable to those of RB in diluted solution, displaying modest fluorescence and high singlet oxygen production, indicative of minimal dye aggregation.

While the surface density of RB molecules was maintained across all nanoparticle sizes, the overall RB loading per gram of nanoparticles decreased with increasing particle size. The 250 nm nanoparticles exhibited an order-of-magnitude lower RB loading compared to the 20 nm system, rendering the larger particles less effective for aPDT applications. Our bacterial assays demonstrated that the use of silica nanoparticles as carriers effectively minimized the intrinsic cytotoxicity of RB in the dark while enabling significant phototoxic activity under illumination. The addition of the gluconamide targeting moiety further enhanced photodynamic performance by promoting selective interaction with bacterial membranes. Among the tested systems, the smallest nanoparticles, RB-G-20@SiNP, exhibited the highest phototoxicity-to-cytotoxicity ratio, closely followed by the intermediate-sized RB-G-80@SiNP.

The covalent attachment of RB molecules to the nanoparticle surface ensured their immobilization, preventing diffusion into the surrounding medium and eliminating spill-over effects associated with free photosensitizers such as Rose Bengal, which can unintentionally interact with healthy tissues. The incorporation of gluconamide as a bacterial ligand facilitated selective targeting of bacterial membranes, creating a dual strategy that combines photosensitizer immobilization with bacterial targeting. This approach ensures precise photodynamic activity, minimizes off-target effects [57], and maintains efficacy even at lower photosensitizer concentrations. By addressing both the spatial confinement of singlet oxygen generation and the biological specificity of the photosensitizer, the developed nanosystems provide a significant advantage over nonspecific photosensitizer delivery.

Overall, our study underscores the potential of small silica nanoparticles (20 nm and 80 nm) functionalized with RB and gluconamide as highly effective nanoplatforms for aPDT, achieving enhanced photodynamic outcomes with minimal dark toxicity. These nanosystems represent a safe and efficient approach for photoactivated antimicrobial treatment. Future research will explore the incorporation of alternative photosensitizers to further expand the therapeutic potential and adaptability of this nanotechnology in combating antimicrobial resistance. Particular attention will be given to red-absorbing photosensitizers, which offer advantages such as improved light penetration into biological tissues and minimized off-target effects, making them more suitable for clinical applications.

## Figures and Tables

**Figure 1 nanomaterials-14-01982-f001:**
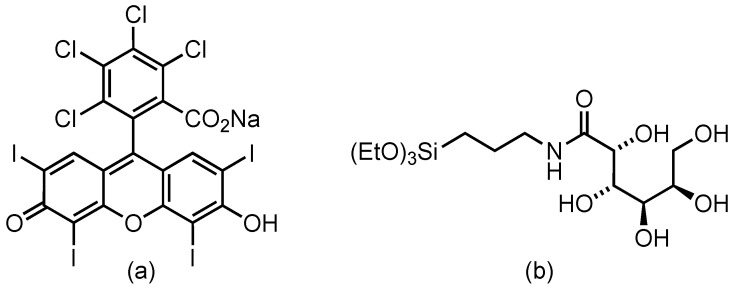
Molecular structures of Rose Bengal (**a**) and silylated gluconamide derivative (**b**).

**Figure 2 nanomaterials-14-01982-f002:**
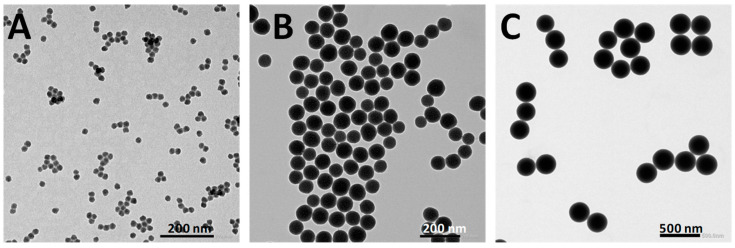
TEM images of 20@SiNPs, 80@SiNPs, and 250@SiNPs suspended in water (scale bar 200 nm in (**A**,**B**) and 500 nm in (**C**)).

**Figure 3 nanomaterials-14-01982-f003:**
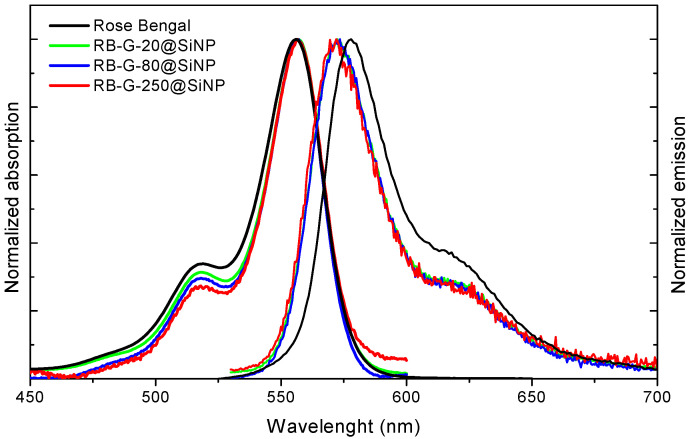
Normalized absorption and emission spectra of RB (black), RB-G-20@SiNP (green), RB-G-80@SiNP (blue), and RB-G-250@SiNP (red) in methanol.

**Figure 4 nanomaterials-14-01982-f004:**
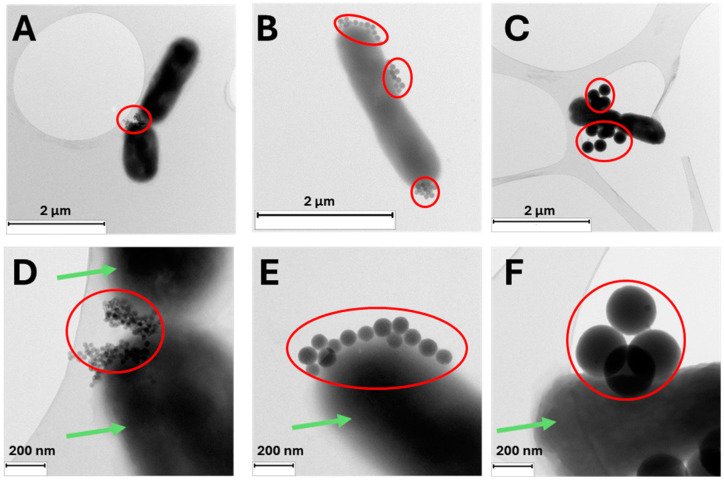
Representative TEM images of a fixed *E. coli* bacteria (green arrow) exposed to RB-G-20@SiNP (**A**,**D**), RB-G-80@SiNP (**B**,**E**), and RB-G-250@SiNP (**C**,**F**) at 0.15 mg/mL (red circle). The samples were deposited on carbon-coated copper grids.

**Figure 5 nanomaterials-14-01982-f005:**
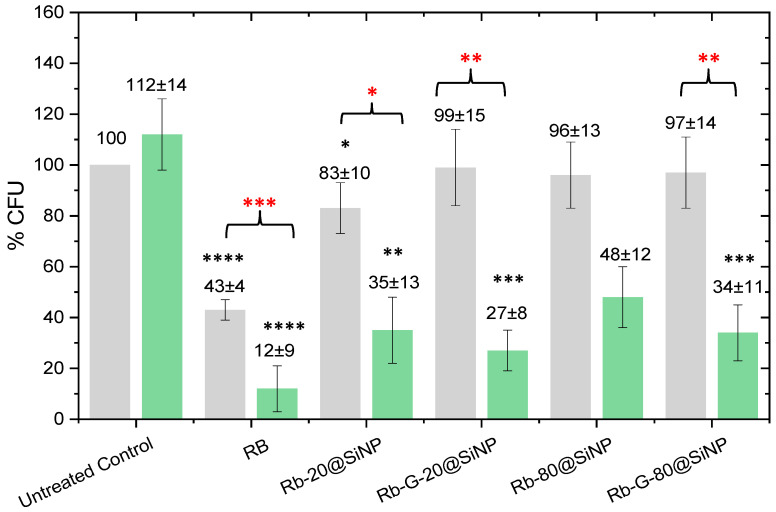
CFU % (as viability test) of *E. coli* bacteria in dark (grey bar) and under 2 h of green irradiation (green bar) incubated with samples normalized at 5 × 10^−7^ M concentration of RB. Black asterisks indicate a significant difference compared to controls. Red asterisks indicate significant differences between dark and light conditions in the same sample tested. *p* value: * < 0.033, ** < 0.002, *** < 0.0002, and **** < 0.0001, n = 3, DF = 4.

**Table 1 nanomaterials-14-01982-t001:** Silica nanoparticle size by TEM and dynamic light scattering (DLS), ζ-potential (ζ), and surface atomic composition (XPS) for the different samples.

Samples	TEM (nm)	DLS (nm)	PDI Index	ζ (mV)	XPS		
					%C	%O	%Si
20@SiNP	21 ± 2	50	0.44	−31	2.8	68.8	28.3
80@SiNP	80 ± 5	100	0.21	−33	3.1	63.0	33.9
250@SiNP	260 ± 27	250	0.20	−53	2.0	70.1	27.9

**Table 2 nanomaterials-14-01982-t002:** Dynamic light scattering (DLS) size, PDI index and ζ-potential (ζ), and RB loading for the different nanosystems.

Samples	DLS (nm)	PDI Index	ζ (mV)	(RB) μmol/g
RB-20@SiNP	41	0.37	−22	7.0
RB-G-20@SiNP	45	0.44	−29	7.0
RB-80@SiNP	81	0.14	−23	2.1
RB-G-80@SiNP	83	0.25	−40	2.1
RB-250@SiNP	261	0.17	−44	0.8
RB-G-250@SiNP	280	0.25	−46	0.8

**Table 3 nanomaterials-14-01982-t003:** Photophysical properties of the complete samples: absorption (λ_ab_) and fluorescence (λ_fl_) wavelength, and fluorescence (Φ_fl_), and singlet oxygen (Φ_∆_) quantum yield.

Samples	λ_ab_ (nm)	λ_fl_ (nm)	Φ_fl_	Φ_∆_
RB	556	578	0.10	0.86
RB-G-20@SiNP	557	572	0.07	0.84
RB-G-80@SiNP	557	573	0.05	0.76
RB-G-250@SiNP	557	576	0.02	0.70

## Data Availability

The raw data supporting the conclusions of this article will be made available by the authors on request.

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
