# Peer review of "Exploring Gluconamide-Modified Silica Nanoparticles of Different Sizes as Effective Carriers for Antimicrobial Photodynamic Therapy"

_nanomaterials, 2024, doi:10.3390/nano14241982_

Round 1

Reviewer 1 Report

Comments and Suggestions for Authors

In this short MS silicon nanoparticles are surface functionalized with gluconamide for affinity-targeting to Gramm negative bacteria, and with Rose Bengal as photosensitizer. The authors examined nano-particles of three sizes (20, 80 250 nm), found that smaller nanoparticles (20, 80 nm), with simultaneous conjugation of gluconamide and Rose Bengal, are likely more photodynamically toxic.

There are three mis-conceptions that need to be addressed.

1/ Please consider Rose Bengal density per surface area, instead of dye / nanoparticle weight ratio. 

2/ Degree of statistical analysis does not correlate with degree of phototoxicity, the mean toxicity (% CFU in Figure 5) does. Please correct this throughout the text!

3/ This work does not contain a Discussion Section.

Please change the Section of Conclusion to Discussion! Currently only the last Para is Conclusion.

In Discussion, the authors might want to:

(a) Discuss about lower concentrations of Rose Bengal in solution with no dark toxicity. PDT with large dark toxicity does not convince!

(b) Stress the dis-advantages of spill-over effects of non-specifically-localized photosensitizers - such as free Rose Bengal, and nano-particles without the gluconamide conjugation. It is known that low doses of PDT could actually trigger intrinsic physiological responses (Cui ZJ, Kanno T, J Physiol, 1997) instead of killing cells. 

(c) Discuss the protein corona effect?

Other points

4/ Too many subtitles in Methods. Suggestions: Try to use two-grade subtitles, for example, remove the numbers of 2.2.1 - 2.2.4 from the subtitle.  

5/ Places that needs revisions.

Lines 60,61: Language needs polishing: Should be "As the action of PDT", "infected area".

Line 80: Delete either "size" or "diameters".

Line 91: Please add a Ref. for "protein corona formation", for those readers who are not familiar with this term.

Line 134: Delete "mixture was".

Line 136 and elsewhere: Please give the centrifugal force in units of g.

Line 199: Scheme S23?

Line 206: Whilst it is good to know the exact P values, for statistical significance, a P value of < 0.05 is sufficient!

Line 285: Remove the question mark.

Lines 310-312: The meaning of this sentence is unclear. How do you mean by incubation, incubation with the photosensitizers?

Line 316: The larger "grade of significance" does not mean larger effect!

Line 363: "RB loading on surface decreased". This may not be true. Please calculate surface area density of RB to be sure. 

You mentioned GFP, but GFP-E. coli not used in the Results Section?

6/ Please give the power density of the LED 518 nm light source in mWatt/cm2. 

7/ Results, Figure 5: Please reduce the concentration of the photosensitizers used, especially for Rose Bengal, to close to zero dark toxicity, then examine the photokilling effect, for each of the photosensitizers (Rose Bengal, RB-20@SiNP, RB-80@SiNp, RB-G-20@SiNP, RB-G-80@SiNP). 

8/ In Discussion, please discuss the surface Rose Bengal concentration for each of the nanoparticles used, and compare those with Rose Bengal concentration in solution.   

Author Response

In this short MS silicon nanoparticles are surface functionalized with gluconamide for affinity-targeting to Gramm negative bacteria, and with Rose Bengal as photosensitizer. The authors examined nano-particles of three sizes (20, 80 250 nm), found that smaller nanoparticles (20, 80 nm), with simultaneous conjugation of gluconamide and Rose Bengal, are likely more photodynamically toxic.

There are three mis-conceptions that need to be addressed.

1/ Please consider Rose Bengal density per surface area, instead of dye / nanoparticle weight ratio. 

Reply. We thank the reviewer for this valuable comment. The number of RB molecules per nanoparticles and per surface unit (nm2) is now calculated (see new Table S1 in ESI) and the results are described in the text.  

2/ Degree of statistical analysis does not correlate with degree of phototoxicity, the mean toxicity (% CFU in Figure 5) does. Please correct this throughout the text!

Reply. We apologize for the misunderstanding. In the revised version, we have accordingly modified the sentences in which the significance degree in the statistical analysis was related to the phototoxicity efficiency.

3/ This work does not contain a Discussion Section.

Please change the Section of Conclusion to Discussion! Currently only the last Para is Conclusion.

Reply. As the reviewer 3 suggested, we have joined both sections into one, renamed as Results and Discussion. We consider more appropriate the analysis of the results as they are being cited along the text.

As the reviewer suggests, part of the conclusions is now moved to “Results and Discussion” section.

In Discussion, the authors might want to:

(a) Discuss about lower concentrations of Rose Bengal in solution with no dark toxicity. PDT with large dark toxicity does not convince!

(b) Stress the dis-advantages of spill-over effects of non-specifically-localized photosensitizers - such as free Rose Bengal, and nano-particles without the gluconamide conjugation. It is known that low doses of PDT could actually trigger intrinsic physiological responses (Cui ZJ, Kanno T, J Physiol, 1997) instead of killing cells. 

(c) Discuss the protein corona effect?

Reply: Most of these suggestions have been taken into account in the revised version of the manuscript. All new additions in the text are now highlighted in yellow.

Briefly, the RB concentration and light doses were already optimized in previous work (mentioned in experimental). RB has been shown to be cytotoxic even at very low concentrations (5 10-7M). The origin of this cytotoxicity is not yet fully understood. This point is already mentioned in the main text. However, we consider that further reduction of the concentration (<10-7 M) is not advisable. Actually, the use of nanoparticles allows the incubation of larger amount of PS without activating dark toxicity, which is one of the main objectives of the present work: minimizing cytotoxicity in the dark while attaining efficient phototoxicity action.

We agree with the reviewer that non-specifically- localized PS could cause side effects. However, considering the short half-life of singlet oxygen in cells (< 0.05 μs), its diffusion is limited to very small distances, thereby reducing the probability of unintended damage. For this reason, targeting remains a critical aspect of PDT.

To address this concern, we have added a comprehensive explanation in the main text about the corona effect on the targeting ability of the nanosystem to make it clearer for the reader. We have included two relevant references to support this discussion.

Other points

4/ Too many subtitles in Methods. Suggestions: Try to use two-grade subtitles, for example, remove the numbers of 2.2.1 - 2.2.4 from the subtitle.  

Reply: They have been removed, as suggested.

5/ Places that needs revisions.

Lines 60,61: Language needs polishing: Should be "As the action of PDT", "infected area".

Reply: revised.

Line 80: Delete either "size" or "diameters".

Reply: deleted size, as suggested.

Line 91: Please add a Ref. for "protein corona formation", for those readers who are not familiar with this term.

Reply: two new references have been added, one related to the general concept of protein corona formation and another specifically addressing the effect of corona formation on silica nanoparticles.

Line 134: Delete "mixture was".

Reply: In has been deleted.

Line 136 and elsewhere: Please give the centrifugal force in units of g.

Reply: The unit rcf (Relative Centrifugal Force) is equivalent to g force and is included in the text alongside rpm.

Line 199: Scheme S23?

Reply: We apologize for the mistake, Scheme S2 is now corrected

Line 206: Whilst it is good to know the exact P values, for statistical significance, a P value of < 0.05 is sufficient!

Reply: we thank the reviewer for the comment but as the statistical is done, we decided to include it.

Line 285: Remove the question mark.

Reply: I has been deleted.

Lines 310-312: The meaning of this sentence is unclear. How do you mean by incubation, incubation with the photosensitizers?

Reply: Sentence rewritten to make it clearer. We were referring to the irradiation time, not the incubation time)

Line 316: The larger "grade of significance" does not mean larger effect!

Reply: we agree, we have modified the text accordingly.

Line 363: "RB loading on surface decreased". This may not be true. Please calculate surface area density of RB to be sure. 

Reply: The reviewer is right. After the calculation (see Table S1 in ESI), similar number of RB molecules are attached per nm2 of external surface of silica nanoparticles. We have included this result in the main text. Even now, RB amount per gram of NP is in agreement with the surface-area-to-volume ratio, which increase as particle size decrease.

You mentioned GFP, but GFP-E. coli not used in the Results Section?

Reply: Thank you for your observation regarding GFP-E. coli. You are absolutely correct, and we appreciate you bringing this to our attention. GFP-E. coli was not used in the Results Section, and its mention in the manuscript was an oversight. We will remove this reference in the revised version to ensure clarity and accuracy. We are grateful for your valuable feedback.

Please give the power density of the LED 518 nm light source in mWatt/cm2. 

Reply: The power density is added in the main text (10mW/cm2)

7/ Results, Figure 5: Please reduce the concentration of the photosensitizers used, especially for Rose Bengal, to close to zero dark toxicity, then examine the photokilling effect, for each of the photosensitizers (Rose Bengal, RB-20@SiNP, RB-80@SiNp, RB-G-20@SiNP, RB-G-80@SiNP). 

8/ In Discussion, please discuss the surface Rose Bengal concentration for each of the nanoparticles used, and compare those with Rose Bengal concentration in solution.   

Reply: As the reviewer suggested, a similar number of RB molecules per surface area of nanoparticles is attached. However, the best PDT performance is assigned to a higher RB amount per gram of NP, which is fulfilled by the smallest particle size of this series (in accordance with the highest surface-area-to-volume ratio). The bioassays were done using the same RB concentration in terms of molarity (mol/L), both in solution and when attached to the surface, to make them comparable.

Reviewer 2 Report

Comments and Suggestions for Authors

The manuscript entitled "Exploring Gluconamide-Modified Silica Nanoparticles of Different Sizes as Effective Carriers for Antimicrobial Photodynamic Therapy" describes the synthesis and characterization of Gluconamide-modified silica nanoparticles and their application in antimicrobial photodynamic therapy. The modified SiO₂ nanoparticles of different sizes (20, 80, and 250 nm) were characterized using dynamic light scattering (DLS), photophysical analysis, transmission electron microscopy (TEM), and X-ray photoelectron spectroscopy (XPS). The antimicrobial activity of the nanoparticles against Gram-negative bacteria, such as E. coli, was evaluated. The effect of nanoparticle size on antimicrobial activity was investigated, and it was demonstrated that smaller nanoparticles (20 nm) exhibited better performance than larger nanoparticles (80 nm).

The following comments should be addressed before considering the manuscript for publication in Nanomaterials:

  • In Scheme 1.1, the functionality of Gluconamide is missing. The authors should reconsider the representation of the covalent attachment of Rose Bengal and Gluconamide moieties to the surface of the silica nanoparticles.
  • FT-IR spectroscopy should be used to demonstrate the covalent attachment of the functional groups.
  • The amount of Rose Bengal and Gluconamide attached could be supported by thermogravimetric analysis (TGA).
  • An explanation is required for why the 80 nm SiNPs exhibited higher percentages of oxygen (O) and silicon (Si) than expected.
Comments on the Quality of English Language

The manuscript entitled "Exploring Gluconamide-Modified Silica Nanoparticles of Different Sizes as Effective Carriers for Antimicrobial Photodynamic Therapy" describes the synthesis and characterization of Gluconamide-modified silica nanoparticles and their application in antimicrobial photodynamic therapy. The modified SiO₂ nanoparticles of different sizes (20, 80, and 250 nm) were characterized using dynamic light scattering (DLS), photophysical analysis, transmission electron microscopy (TEM), and X-ray photoelectron spectroscopy (XPS). The antimicrobial activity of the nanoparticles against Gram-negative bacteria, such as E. coli, was evaluated. The effect of nanoparticle size on antimicrobial activity was investigated, and it was demonstrated that smaller nanoparticles (20 nm) exhibited better performance than larger nanoparticles (80 nm).

The following comments should be addressed before considering the manuscript for publication in Nanomaterials:

  • In Scheme 1.1, the functionality of Gluconamide is missing. The authors should reconsider the representation of the covalent attachment of Rose Bengal and Gluconamide moieties to the surface of the silica nanoparticles.
  • FT-IR spectroscopy should be used to demonstrate the covalent attachment of the functional groups.
  • The amount of Rose Bengal and Gluconamide attached could be supported by thermogravimetric analysis (TGA).
  • An explanation is required for why the 80 nm SiNPs exhibited higher percentages of oxygen (O) and silicon (Si) than expected.

Author Response

Comments and Suggestions for Authors

The manuscript entitled "Exploring Gluconamide-Modified Silica Nanoparticles of Different Sizes as Effective Carriers for Antimicrobial Photodynamic Therapy" describes the synthesis and characterization of Gluconamide-modified silica nanoparticles and their application in antimicrobial photodynamic therapy. The modified SiO₂ nanoparticles of different sizes (20, 80, and 250 nm) were characterized using dynamic light scattering (DLS), photophysical analysis, transmission electron microscopy (TEM), and X-ray photoelectron spectroscopy (XPS). The antimicrobial activity of the nanoparticles against Gram-negative bacteria, such as E. coli, was evaluated. The effect of nanoparticle size on antimicrobial activity was investigated, and it was demonstrated that smaller nanoparticles (20 nm) exhibited better performance than larger nanoparticles (80 nm).

The following comments should be addressed before considering the manuscript for publication in Nanomaterials:

1/ In Scheme 1.1, the functionality of Gluconamide is missing. The authors should reconsider the representation of the covalent attachment of Rose Bengal and Gluconamide moieties to the surface of the silica nanoparticles.

Reply: Scheme 1.1 has been changed. Now the RB and Gluconamide attachment are depicted. 

2/FT-IR spectroscopy should be used to demonstrate the covalent attachment of the functional groups.

Reply: FTIR spectra have now been performed for the bare and functionalized silica nanoparticles. Characteristic peaks of amide groups (at ~1635 cm–1 and ~1550 cm–1) are observed in the spectra of the functionalized systems, confirming the covalent attached of RB and G at the external surface of the nanoparticles. The spectra are included in Figure S6 in the ESI, and the results are described in the main text. We have also include the FTIR technique specifications in the experimental section.

3/The amount of Rose Bengal and Gluconamide attached could be supported by thermogravimetric analysis (TGA).

Reply: The most accurate method for estimating the RB loading is the photometric method. It provides quantitative data whereas TG can only offer qualitative information. Besides, the organic amount is low relative to silica content (< 5% in weight ratio) and TG is not considered a suitable technique for this purpose.

4/An explanation is required for why the 80 nm SiNPs exhibited higher percentages of oxygen (O) and silicon (Si) than expected.

Reply: In this sample, %Si is slightly higher, while %O is slightly lower compared to the other samples, but the Si:O ratio remains close to 1:2. In fact, this sample follows a Si:O ratio even closer to 1:2 than the 20 nm and 250 nm samples. The slight variation can be a consequence of resolution limits.

Reviewer 3 Report

Comments and Suggestions for Authors

The authors prepared silica nanoparticles of different sizes which were functionalized with the photosensitizer Rose Bengal (RB) and a gluconamide ligand for the purpose of generating ROS and increasing selectivity, respectively. Interesting results were obtained. However, methodological doubts arise after reviewing the manuscript, so it is important to address the comments before publishing:

In the introduction they focus on bacterial resistance but a resistant strain is not included in the design. Include how the results could impact bacterial resistance

line 137: How was the ethanol completely removed? Starting from the fact that it has antimicrobial properties?

Line 181: How were these strains confirmed? What is the reference strain? What was the plasmid used? How were they transformed?

Line 193: But GFP is not excited at this wavelength, it is not entirely clear why recombinant bacteria were used. The purpose was to excite the GFP to emit at a wavelength for the photosensitizer? Please clarify

Line 206: Describe the statistical analysis more precisely

Line 216: "and discussion"

Line 221: Could PDI confirm this statement

Line 285: Why the question?

Figure 4. TEM illustrates a system that preserves the integrity of nanoparticles, in fact there are many that do not adhere to the bacterial surface. The images are 2 hours later, that is, after the radiation where the ROS should have destroyed the bacteria. What happens to the ROS that do not come into contact with the surface? Do they represent any risk of cytotoxicity for a potential host despite the selectivity for gluconamide? Discuss

Line 322: Releases ROS by the emission of GFP that has not been excited? Please clarify

Line 342: How did it overcome the outer membrane barrier? Please propose a model

Author Response

The authors prepared silica nanoparticles of different sizes which were functionalized with the photosensitizer Rose Bengal (RB) and a gluconamide ligand for the purpose of generating ROS and increasing selectivity, respectively. Interesting results were obtained. However, methodological doubts arise after reviewing the manuscript, so it is important to address the comments before publishing:

1/line 137: How was the ethanol completely removed? Starting from the fact that it has antimicrobial properties?

Reply: After the centrifugation process to remove the excess of non-reactive dye, the nanoparticles were rinsed with water, performing several rinsing steps to ensure the complete removal of ethanol. Indeed, the nanoparticles show no cytotoxic effects in the dark.

2/In the introduction they focus on bacterial resistance but a resistant strain is not included in the design. Include how the results could impact bacterial resistance

You mentioned GFP, but GFP-E. coli not used in the Results Section?

Reply: Thank you for your observation regarding GFP-E. coli. You are absolutely correct, and we appreciate you bringing this to our attention. GFP-E. coli was not used in the Results Section, and its mention in the manuscript was an oversight. We have removed this reference in the revised version to ensure clarity and accuracy. We are grateful for your valuable feedback.

3/In the introduction they focus on bacterial resistance but a resistant strain is not included in the design. Include how the results could impact bacterial resistance

Reply: Thank you for your insightful comment. We would like to clarify that the E. coli strain used in our study is resistant to chloramphenicol and ampicillin. However, this work serves as a proof of concept, aiming to establish the foundational methodology. The best-performing system obtained here will be further tested in more resistant bacterial strains in future studies to better address the challenges posed by bacterial resistance.

4/Line 181: How were these strains confirmed? What is the reference strain? What was the plasmid used? How were they transformed?

Reply: Thank you for your questions. The strain used in this study is E. coli BL21(DE3), a derivative of the E. coli B strain, which is widely used for recombinant protein expression. The BL21(DE3) strain was first described by Studier and Moffatt in 1986.

5/Line 193: But GFP is not excited at this wavelength, it is not entirely clear why recombinant bacteria were used. The purpose was to excite the GFP to emit at a wavelength for the photosensitizer? Please clarify

Reply: We sincerely apologize for the confusion caused by the inclusion of GFP-E. Coli in the manuscript. The reference to GFP-E. Coli in the Materials and Methods section was inadvertently carried over from other related publications during manuscript preparation. We regret this oversight and will ensure it is corrected in the revised version. Thank you for highlighting this point, and we appreciate your careful review.

6/Line 206: Describe the statistical analysis more precisely

Reply: A more detailed description of the statistical analysis is now added to the experimental section.

7/Line 216: "and discussion"

Reply: As the reviewer suggests that section is renamed as Results and discussion

8/Line 221: Could PDI confirm this statement

Reply: The PDI values are now included in table 2. The PDI values ≤ 0.25 obtained for functionalized nanoparticles with 80 nm and 250 nm diameters indicate a narrow size distribution whereas the slightly larger (~0.4) PDI value for the 20 nm nanoparticles suggests a broader size distribution. This information is now included in the revised version.

9/Line 285: Why the question?

Reply: We apologize for this mistake, the question mark has been removed.

10/Figure 4. TEM illustrates a system that preserves the integrity of nanoparticles, in fact there are many that do not adhere to the bacterial surface. The images are 2 hours later, that is, after the radiation where the ROS should have destroyed the bacteria.

Reply: TEM images has been taken just to illustrate the interaction of the surface nanoparticles with the bacterial cell wall. In these images, the samples were not irradiated, the nanoparticles were just incubated together with the bacteria. Therefore, no toxic effects are expected, as the nanoparticles are not cytotoxic under dark conditions.

11/What happens to the ROS that do not come into contact with the surface? Do they represent any risk of cytotoxicity for a potential host despite the selectivity for gluconamide? Discuss

Reply: PDT is considered a localized treatment as ROS are produced only at the irradiated area. In fact, the diffusion of ROS (singlet oxygen) is very limited, due to its very short lifetime <0.04 us, making it unlikely to affect distant targets. Therefore, only the nanoparticles located at the bacterial cell wall can disrupt it following light irradiation.

12/Line 322: Releases ROS by the emission of GFP that has not been excited? Please clarify

13/Line 342: How did it overcome the outer membrane barrier? Please propose a model

Reply: the presence of gluconamide as a ligand on the silica surface promotes the interaction with the lipopolissacharde chains on the outer membrane of the bacteria, based on a previous study (ref Capeletti, L.B.; de Oliveira, J.F.A.; Loiola, L.M.D.; Galdino, F.E.; da Silva Santos, D.E.; Soares, T.A.; de Oliveira Freitas, R.; Cardoso, M.B. Gram-Negative Bacteria Targeting Mediated by Carbohydrate–Carbohydrate Interactions Induced by Surface-Modified Nanoparticles. Advanced Functional Materials 2019, 29, 1–11). The nanoparticles do not need to be internalized by the bacteria, they only need to localize at the membrane. However, singlet oxygen, due to its small molecular size, can penetrate the bacterial membrane after the light irradiation (check this reference: T.A. Dahl, W. R. Midden, P.E. Hartman Journal of Bacteriology, 1989, Vol. 171, No. 4 p. 2188-2194).

Round 2

Reviewer 1 Report

Comments and Suggestions for Authors

The revised MS is improved but mis-conceptions and poor presentations remain.  

Major points

1/ Even if the nanoparticle surface Rose Bengal concentrations remain identical with varied size of the nanoparticles, the authors still wish to stress the misleading idea of Rose Bengal weight / nanoparticle weight ratios. Only surface concentration counts! You need to stress the differences in contact-area between bacteria and nano-particle, not the weight ratios! 

2/ Regarding the photosensitizer specificity for Discussion, the authors did not get the point (which the Referee pointed out before). With higher concentrations of photosensitizers, PDT may kill the targeted cells, but without specific targeting of photosensitizers, lower concentration may spread to un-intended normal tissue in PDT, therefore leading to lowered doses of PDT there, not to kill the un-intended cells, but to activate normal physiology such as calcium oscillations. The authors need to make this point clear to the readers to strengthen their argument for specific targeting, by conjugation to gluconamide, for example. Although the idea of limited diffusion distance of singlet oxygen has changed slightly during the last decades, to quote the short diffusion distance of singlet oxygen is off the point here! 

3/ The authors claim that Rose Bengal has intrinsically high dark toxicity, therefore the effective phototoxic concentration must remain high (at 0.5 microM). Here clearly the authors made a bad choice (with > 50% dark toxicity, see Figure 5) by using Rose Bengal. Any PDT effect is meaningless with significant dark toxicity. You need to use lower Rose Bengal concentrations (such as 100 nM) where no or minimum dark toxicity is observed! Then compare this with the nanoparticle surface concentration. I would suggest that a better photosensitizer be added in this work for comparison.  A great many photosensitizers are available which have low dark toxicity. This includes candidates on the FDA (of many countries) approved list. The green light-absorbing Rose Bengal is not ideal in terms of light penetration into biological tissues, anyway. Red-absorbing photosensitizers would be more suited for in vivo applications. If you could not do a new photosensitizer, at least discuss about it in the Section for Discussion. 

4/ Combining Conclusion with Results to be the new Results + Discussion Section does not change that fact that Discussion of the MS is far from adequate! 

5/ In a new Section for Discussion, you could discuss at least: (a) The geometry effects of nanoparticles of the three different sizes. (b) The advantages of photosensitizer specificity, including specificity afforded by conjugated gluconamide. (c) The un-wanted side effects of diffused photosensitizer to surrounding normal tissues and interference of physiological functions. (d) The disadvantages of the corona effect and some other points. 

Minor points

6/ The title could be changed: To Optimize the Size of Gluconamide-Modified Silica Nanoparticle as Effective Carrier for Antimicrobial Photodynamic Therapy.  

7/ Lines 24-27: Surface loading density of Rose Bengal was constant, but the spherical geometry of nanoparticles favored smaller nanoparticles for antibacterial PDT. 

8/ Please replace the colloquial "Besides" with the more formal "Further,".

9/ In Introduction, some more detailed description of the "corona formation" is called for. 

10/ Line 198: Change to "518 nm, 10 mW/cm2". 

11/ Line 304: Since the surface density is similar, to say "low RB loading" is probably incorrect. 

12/ Figure 4. Please describe the cell culture matrix (carbon-coated copper grids?) in the legend. Further, I suggest that you add arrows to indicate nano-particles and bacteria. 

13/ Lines 325: Should be "light doses". 

14/ Refs. No. 53, 54 are incomplete. 

Comments on the Quality of English Language

Could be improved. 

Author Response

Reply: The following comments, together with the described associated changes introduced in the manuscript, are intended to address the remaining concerns expressed by Reviewer 1 on some aspects of the wok.

The revised MS is improved but mis-conceptions and poor presentations remain.  

Major points

1/ Even if the nanoparticle surface Rose Bengal concentrations remain identical with varied size of the nanoparticles, the authors still wish to stress the misleading idea of Rose Bengal weight / nanoparticle weight ratios. Only surface concentration counts! You need to stress the differences in contact-area between bacteria and nano-particle, not the weight ratios! 

Reply: Thank you for your insightful comments. As you have pointed out, we have kept the surface density of RB molecules constant across all nanoparticle sizes. We would like to clarify that, despite this constant surface density, the size of the nanoparticles can still influence their photodynamic antibacterial activity. Smaller nanoparticles, with their higher surface-to-volume ratio, present a larger number of nanoparticles per unit volume, which increases their contact with the bacterial cell membrane. This results in a higher effective generation of singlet oxygen and potentially more efficient antimicrobial activity. Therefore, smaller nanoparticles (e.g., 20 nm and 80 nm) are expected to show improved antibacterial performance relative to the larger 250 nm nanoparticles, despite all particles having the same RB surface density. We hope this explanation resolves your query, and we have included this discussion in the revised manuscript. To provide further context and support for our discussion, we have included two new references from the literature that explore the relationship between nanoparticle size, surface interactions, and antimicrobial efficacy.

This clearly explains how particle size, even when surface density and total concentration of RB are controlled, impacts the effectiveness of the treatment due to the physical properties of the nanoparticles.

2/ Regarding the photosensitizer specificity for Discussion, the authors did not get the point (which the Referee pointed out before). With higher concentrations of photosensitizers, PDT may kill the targeted cells, but without specific targeting of photosensitizers, lower concentration may spread to un-intended normal tissue in PDT, therefore leading to lowered doses of PDT there, not to kill the un-intended cells, but to activate normal physiology such as calcium oscillations. The authors need to make this point clear to the readers to strengthen their argument for specific targeting, by conjugation to gluconamide, for example. Although the idea of limited diffusion distance of singlet oxygen has changed slightly during the last decades, to quote the short diffusion distance of singlet oxygen is off the point here! 

Reply: Thank you for your constructive comments, which help to clarify key aspects of our work. We emphasize that the RB molecules in our system are covalently attached to the silica nanoparticles, ensuring they are immobilized and preventing their diffusion into surrounding tissues or unintended cells. This immobilization eliminates the possibility of spill-over effects caused by free photosensitizers, such as free RB. Furthermore, the functionalization of the nanoparticles with the gluconamide ligand, provides specificity for bacterial membranes, reducing the likelihood of the photosensitizer interacting with healthy tissues. This specificity is a critical advancement, as it minimizes off-target effects and ensures the photodynamic activity is localized to the bacterial cells. We appreciate your comment regarding the limited diffusion distance of singlet oxygen. While singlet oxygen's short half-life reduces its capacity to cause off-target damage, we acknowledge that this alone does not fully address concerns regarding photosensitizer localization. By combining immobilized photosensitizers with a bacterial-targeting ligand, our system addresses both spatial confinement and biological specificity. We have introduced new text into the conclusions section to better clarify this point, including the new reference describing photodynamic off-target effects.

Finally, as discussed in our previous responses, the protein corona effect has been addressed in the revised manuscript with a detailed explanation and relevant references.

3/ The authors claim that Rose Bengal has intrinsically high dark toxicity, therefore the effective phototoxic concentration must remain high (at 0.5 microM). Here clearly the authors made a bad choice (with > 50% dark toxicity, see Figure 5) by using Rose Bengal. Any PDT effect is meaningless with significant dark toxicity. You need to use lower Rose Bengal concentrations (such as 100 nM) where no or minimum dark toxicity is observed! Then compare this with the nanoparticle surface concentration. I would suggest that a better photosensitizer be added in this work for comparison.  A great many photosensitizers are available which have low dark toxicity. This includes candidates on the FDA (of many countries) approved list. The green light-absorbing Rose Bengal is not ideal in terms of light penetration into biological tissues, anyway. Red-absorbing photosensitizers would be more suited for in vivo applications. If you could not do a new photosensitizer, at least discuss about it in the Section for Discussion. 

Reply: We appreciate the opportunity to clarify our approach and address your concerns.

Rose Bengal (RB) was chosen for this study as a well-established photosensitizer model due to its high singlet oxygen quantum yield and straightforward functionalization onto silica nanoparticles. While we acknowledge the intrinsic dark toxicity of free RB at concentrations above 0.5 µM, our work demonstrates that conjugating RB to silica nanoparticles significantly reduces dark toxicity compared to the free dye. This key feature highlights the advantage of nanoparticle-based systems in minimizing off-target effects while preserving high phototoxicity under irradiation, as explain in the newly revised manuscript.

Regarding the suggestion to explore lower RB concentrations (e.g., 100 nM), we have previously optimized the RB loading and light exposure conditions to balance photodynamic efficiency and dark cytotoxicity, as explained in the manuscript. Concentrations below this threshold showed suboptimal phototoxic effects in our assays. Furthermore, the use of FDA-approved photosensitizers, while scientifically interesting, falls outside the current scope of this study. Such an extension would require extensive optimization and comparative studies that would shift the focus from our primary objective of evaluating the nanosystem's functionality and targeting capabilities.

Of course, we agree that red-absorbing photosensitizers offer advantages for in vivo applications due to better tissue penetration of red light. We have now added a small comment in the Conclusions section acknowledging this limitation of RB and emphasizing the potential for future studies to evaluate alternative photosensitizers with red-shifted absorption and lower intrinsic dark toxicity. This would expand the versatility of our platform and improve its relevance for clinical applications.

4/ Combining Conclusion with Results to be the new Results + Discussion Section does not change that fact that Discussion of the MS is far from adequate! 

Reply: See our response to point 1 above.

5/ In a new Section for Discussion, you could discuss at least: (a) The geometry effects of nanoparticles of the three different sizes. (b) The advantages of photosensitizer specificity, including specificity afforded by conjugated gluconamide. (c) The un-wanted side effects of diffused photosensitizer to surrounding normal tissues and interference of physiological functions. (d) The disadvantages of the corona effect and some other points. 

Reply: As stated above, we have addressed several of these points in the revised manuscript. Specifically, we have incorporated discussions on (a) the geometry effects of nanoparticles of different sizes, (b) the advantages of photosensitizer specificity provided by conjugated gluconamide, and (c) the potential side effects of diffused photosensitizer on surrounding normal tissues and physiological functions.

Minor points

6/ The title could be changed: To Optimize the Size of Gluconamide-Modified Silica Nanoparticle as Effective Carrier for Antimicrobial Photodynamic Therapy.  

Reply: We appreciate the referee's suggestion. However, we believe the current title accurately reflects the scope of our study, encompassing both the optimization of nanoparticle size and the demonstration of the efficacy of gluconamide-modified silica nanoparticles as carriers for aPDT. Thus, we propose retaining the original title.

7/ Lines 24-27: Surface loading density of Rose Bengal was constant, but the spherical geometry of nanoparticles favored smaller nanoparticles for antibacterial PDT. 

Reply: We have rephrased this sentence in the abstract section. Now both ideas suggested by the reviewer are included

8/ Please replace the colloquial "Besides" with the more formal "Further,".

Reply We have changed Besides by other more proper synonyms

9/ In Introduction, some more detailed description of the "corona formation" is called for. 

Reply: We have defied what corona formation is (lines 93-96) and two relevant references were added in the previous revision

10/ Line 198: Change to "518 nm, 10 mW/cm2". 

Reply: We have changed it as reviewer suggested

11/ Line 304: Since the surface density is similar, to say "low RB loading" is probably incorrect. 

Reply: We have removed this statement and rewritten the sentence (lines 314-316)

12/ Figure 4. Please describe the cell culture matrix (carbon-coated copper grids?) in the legend. Further, I suggest that you add arrows to indicate nano-particles and bacteria. 

Reply: As reviewer recommended we add arrows in the images and the caption has been completed with the used grids,

13/ Lines 325: Should be "light doses". 

Reply: It has been changed

14/ Refs. No. 53, 54 are incomplete. 

Reply: All the references has been now revised

Reviewer 2 Report

Comments and Suggestions for Authors

The manuscript has been revised satisfactorily according to the suggestions and is recommended for publication in Nanomaterials.

Author Response

We thank the reviewer for his/her positive comments on the revised version of the manuscript. 

Reviewer 3 Report

Comments and Suggestions for Authors

The authors have responded to the comments and the manuscript has been substantially improved.

Author Response

We would like to thank the reviewer for his/her positive comments on the revised version of the manuscript. His/her revision have contributed to the improvement of the quality of the present manuscript

Round 3

Reviewer 1 Report

Comments and Suggestions for Authors

1/ "... , the surface density of RB molecules was calculated, yielding approximately 0.023, 0.027, and 0.029 molecules/nm² for RB-G-20@SiNP, RB-G-80@SiNP, and RB-G-250@SiNP, respectively (Table S1)."

It seems that larger nano-particles actually gather higher surface density of Rose Bengal. The readers may want to know whether the differences in RB density (from 0.023, 0.027 to 0.029 molecules per nm2) are statistically significant! 

2/ To pool with the Results Section only highlight the fact that Discussion remains inadequate. 

3/ The extremely high dark toxicity of Rose Bengal of 0.5 microM at > 50% was not discussed. Most of the time, minimum (< 10%) dark toxicity should be shown when working with PDT. 

Author Response

Comment 1/ "... , the surface density of RB molecules was calculated, yielding approximately 0.023, 0.027, and 0.029 molecules/nm² for RB-G-20@SiNP, RB-G-80@SiNP, and RB-G-250@SiNP, respectively (Table S1)."

It seems that larger nano-particles actually gather higher surface density of Rose Bengal. The readers may want to know whether the differences in RB density (from 0.023, 0.027 to 0.029 molecules per nm2) are statistically significant! 

REPLY: The relative error in the surface density arises from two main sources: (1) the spectrophotometric determination of RB loading, which has an estimated uncertainty of ±5%, and (2) the determination of the specific surface area of the nanoparticles, which depends on the particle diameter and is subject to an estimated uncertainty of ±10% due to particle size distribution. Combining these errors, the total relative uncertainty in the calculated surface density is approximately ±11%.

The absolute errors in the surface densities are as follows:

0.023 molecules/nm² → ±0.0026 molecules/nm²

0.027 molecules/nm² → ±0.0030 molecules/nm²

0.029 molecules/nm² → ±0.0033 molecules/nm²

Given these uncertainties, the differences in surface densities between the samples are within the expected error range and are not statistically significant. This indicates that the surface density of Rose Bengal remains relatively constant across the different nanoparticle sizes as commented in the text, with only minor variations that fall within the margin of error.

However, we would like to emphasize the importance of the RB loading per gram of nanoparticles in photodynamic therapy (PDT) applications. Smaller nanoparticles, with their higher RB loading per gram, provide a significant advantage in PDT, as discussed in the main text. Specifically, for a fixed concentration of RB, systems with higher RB loading per gram require less nanoparticle mass to achieve the desired photosensitizer dose, thereby minimizing potential issues such as nanoparticle aggregation, which reduces efficiency.

We have included the estimated errors in Table S1 of the Supporting Information in the revised to address the referee’s concerns.

Comment 2/ To pool with the Results Section only highlight the fact that Discussion remains inadequate. 

REPLY: We understand the reviewer's concern. However, as mentioned in the previous round of revisions, we chose to present the results and discussion together in a single section because we feel this approach makes the text easier to follow and allows readers to immediately see the interpretation of the findings alongside the data. Additionally, to address concerns about clarity and emphasis on the key findings, we have significantly expanded the Conclusions section in the revised manuscript. This updated conclusions now provides a clear and concise summary of the main messages and their implications.

After carefully reviewing the manuscript again, we are confident that the study's objectives, methodologies, and results are presented in a logical and accessible way, and the key points are adequately addressed throughout the text.

Comment 3/ The extremely high dark toxicity of Rose Bengal of 0.5 microM at > 50% was not discussed. Most of the time, minimum (< 10%) dark toxicity should be shown when working with PDT.

REPLY:We appreciate the reviewer bringing up this point again. As mentioned in the manuscript (lines 341–344), the intrinsic cytotoxicity of RB in solution is not yet fully understood. To address this, we included some hypotheses proposed in the literature (ref. 55 and 56), such as the possibility that RB can penetrate cell membranes under certain conditions of incubation time or concentration. However, investigating this aspect in depth is outside the scope of our study.

Our work focuses on highlighting the advantages of using functionalized nanoparticles to improve PDT treatments—specifically, reducing dark toxicity and increasing photoactivity—compared to free RB in solution. The results clearly demonstrate that the phototoxicity-to-cytotoxicity ratio is consistently higher for RB@NPs than for free RB, regardless of nanoparticle size or the presence of the gluconamide ligand.

As mentioned in a previous revision, we did explore lower RB concentrations (e.g., 100 nM), but these resulted in suboptimal phototoxic effects for the nanosystems in our assays. Therefore, we chose the current experimental conditions to provide meaningful comparisons between the free PS and nanoparticle-based systems.

We hope this clarification addresses the reviewer's concern.